# Can Multifunctional Landscapes Become Effective Conservation Strategies? Challenges and Opportunities From a Mexican Case Study

**Julia Ros-Cuéllar [1], Luciana Porter-Bolland [2,\*] and Martha Bonilla-Moheno [3]**

[1]  Institute of Geography and Spatial Planning, Faculté des Lettres, des Sciences Humaines,
des Arts et des Sciences de l'Education, University of Luxembourg, L-4366 Esch-sur-Alzette, Luxembourg;
julia.roscuellar@uni.lu

[2]  Red de Ecología Funcional, Instituto de Ecología, A. C., 91070 Xalapa, Veracruz, Mexico

[3]  Red de Ambiente y Sustentabilidad, Instituto de Ecología, A. C., 91070 Xalapa, Veracruz, Mexico;
martha.bonilla@inecol.mx

\*  Correspondence: luciana.porter@inecol.mx; Tel.: +52-(228)-842-1800 (ext. 4317)

**Abstract:** Protected Areas (PA) are the main strategy for nature conservation. However, PA are not always efficient for ecological conservation and social wellbeing. A possible alternative for conservation in human-dominated landscapes are Multifunctional Landscapes (ML), which allow the coexistence of multiple objectives, such as nature conservation and resource use. Using the activity system framework, we analyzed whether the ML concept was an operative alternative to PA within an area of interest for conservation in Veracruz, Mexico. Activity systems refer to the set of productive strategies that result from the mobilization of resources and which, within particular environmental governance contexts, shape the landscape. To understand the challenges and opportunities of our case study, we: (1) delimited the landscape according to local conservation interests; and (2) analyzed the role of stakeholders in shaping this landscape. The delimited landscape included areas considered wildlife reservoirs and water provisioning zones. Our results suggested that the existence of local conservation areas (private and communal), combined with shaded-coffee agroforestry practices, made this region an example of ML. Although local conservation initiatives are perceived as more legitimate than top-down approaches, agreements amongst stakeholders are essential to strengthen environmental governance. In specific socio-ecological contexts, ML can be effective strategies for conservation through agroecosystems that maintain a high-quality landscape matrix, allowing nature preservation and delivering economic benefits.

**Keywords:** multifunctional landscape; protected areas; conservation; environmental governance

## 1. Introduction

The designation of Protected Areas (PA), which began almost a century ago, has been the main instrument of biodiversity conservation worldwide [1]. However, the establishment of a PA does not guarantee fulfillment of its goals [2] and it has even been suggested that many of these areas are in reality "paper reserves", since they have no mechanisms by which to ensure effective conservation [3–5]. It is recognized that a large part of this failure is due to the fact that PA policies generally ignore the socio-economic and political context in which they are enacted [6,7]. This has led some authors (i.e., [8,9]) to emphasize the need to implement actions to strengthen environmental governance, understood as the set of formal and informal rules that regulate the interaction between society and nature, and that determine decision-making [10,11]. Strengthening of environmental governance for conservation requires an understanding of local socio-economic and political processes, as well as the

underlying power structures, in order to enable the resolution of conflicts and mediate among the different actors involved in resource use [6,9].

In specific contexts, e.g., in highly populated areas, the necessity of considering alternative conservation strategies to that of PA has been proposed, in which the interests of conservation and those of sustainable natural resource use can converge [5,12,13]. One specific proposal is to promote the establishment of multifunctional landscapes [14], which consider a regional perspective of conservation (the "wider landscape" perspective) in which areas of conservation interest are immersed within a heterogeneous mosaic of land uses and covers that allow the maintenance of ecosystem services and biodiversity, while also satisfying the requirements of social wellbeing [14]. According to Dewi and collaborators (2013) [14], this vision is focused on transcending the "protected vs. non-protected" dichotomy in order to direct efforts towards the management of a matrix in which areas coexist with gradients of conservation and of management with different intensities of use. The proposal is thus focused on the wellbeing of the human population, while ensuring the protection of areas that conserve natural elements [15]. It follows an integrated landscape perspective [16] in which the maintenance of a high-quality matrix allows the preservation of natural patches of vegetation with a high degree of connectedness, sustained by biodiversity-friendly agroecosystems, favoring a land-sharing conservation model rather than a land sparing one [17]. However, for this model to be possible, it is necessary to maintain or develop Sustainable Land Management practices (SLM) within the productive units of the multifunctional landscape. SLM practices require that local resources (e.g., land, soils, water, and plants and animals) be managed "... for the production of goods to meet changing human needs, while simultaneously ensuring the long-term productive potential of these resources and the maintenance of their environmental functions" [18].

This wider landscape vision integrates biophysical, as well as social, political, and economic aspects, in which multiple actors with diverse needs and visions, and of different hierarchical levels, interact [19]. In order to achieve sustainable planning in these contexts, it is necessary to develop strategies that promote agreements among actors with respect to natural resource management [8,19,20]. Therefore, at the core of the proposal of multifunctional landscapes is the development of spaces for appropriate decision-making that, in general, lead to the strengthening of environmental governance as a key factor in the success of conservation and sustainability.

One way to analyze and understand multifunctional landscapes is through the use of the conceptual framework of the activity system [21]. This framework refers to the set of activities that are conducted by a social entity in a specific spatio-temporal context through the mobilization of available resources (e.g., natural, social, financial, material). It is defined by three dimensions: (1) territorial, referring to the geographical space and its characteristics that imply a specific identity created from a shared history and that entail particular knowledge systems; (2) regulatory, referring to the institutions present, together with their current rules and the decision-making spaces; and (3) sectorial, referring to aspects related to policies and the market [21]. This framework thus allows an understanding, from a territorial and socio-ecosystemic perspective, of how the context affects and determines the set of activities and, in turn, how these activities transform the socioecological context in which they occur [21]. A socio-ecosystemic perspective implies considering both the social and the ecological systems as nested, multilevel systems [22] that converge in multiple ways so that its components, such as the resources and units of management, as well as its users and their governance systems, interact, producing outcomes which in turn affect the system and its components [23].

In this study, we contribute to the discussion regarding the development of conservation strategies through planning multifunctional landscapes as an alternative to the establishment of PA. In particular, we analyze the case of the area of interest for conservation known as "Las Cañadas de Sochiapa", in Veracruz, Mexico. In this area, during 2005, a state-level PA was proposed [24]. However, its establishment did not proceed. The initiative failed to a great extent because it was not an agreed request, but rather a situation in which the state governor at the time asked the state counsel for environmental protection in Veracruz to start a fast-track process for the establishment of the PA,

without having produced a solid proposal [24]. Eventually, it was not approved because of a lack of coordination among stakeholders. Soon after that, changes in the local government made the initiative disappear, presumably because it was not reflecting a collective interest or a well-planned regional strategy for nature conservation. However, the area remained as an area of interest for conservation. In this sense, the case presents an interesting context for understanding and discussing the concept of a multifunctional landscape as a conservation strategy, and for the particular case, the strengths and weaknesses in terms of maintaining its properties in the medium- and long-term. We use the conceptual framework of the activity system and the multifunctional landscape approach in the case study in order to (1) delimit the area of interest for conservation as a multifunctional landscape, paying special attention to local interest in conserving this space; (2) understand the role played by the different actors in this landscape; and (3) identify the strengths and weaknesses of this territory in terms of its capacity to maintain its properties in the medium- and long-term. Based on the results, we discuss the possibility of developing strategies that allow scenarios in which multifunctional landscapes can become alternatives to the establishment of PA and catalyze processes towards sustainability.

## 2. Materials and Methods

### 2.1. Study Site

The study region is located in the mid-low part of the watershed of La Antigua river in central Veracruz, Mexico. The zone is fed by more than 20 perennial and intermittent watercourses and the presence of springs [24]. Its elevation ranges from 431 to 1576 masl [25]. The predominant climate is semi-warm humid. It comprises part of the municipalities of Tenampa, Totutla, and Tlaltetela. According to INEGI (2010), the most widespread land use is seasonal agriculture (including coffee plantations; 45%), followed by cultivated grassland (23%) and secondary arboreal vegetation of oak forest (7%). The zone presents a great biological richness and provides multiple environmental services to the region, including water provision, recharging of the aquifer layers, and carbon capture [24]. The zone conserves a large portion of the few relics of tropical low deciduous forest in the state, which is not represented in the existing Protected Areas (PA) [26]. According to the Mexican National Commission for the Knowledge and Use of Biodiversity (CONABIO), the zone is located within the priority terrestrial region 104: "The tropical oak forests of the coastal plain of Veracruz" due to the presence of plant communities considered relics of the Pleistocene [27]. However, the region is vulnerable as a result of agricultural and livestock production activities, which have led to high fragmentation of the original coverage of these oak forests [27].

Due to the importance and biophysical characteristics of the existing natural resources in the zone, in 2005, the Veracruz State Council for Protection of the Environment (COEPA) proposed the establishment of a state-level PA, under the category of a biological corridor, known as "Las Cañadas de Sochiapa" [19,24]. One of its main objectives was to protect the natural springs of this zone and the flows of water they provide to the region, as well as to conserve the prominent gullies it contains. While this proposal did not proceed, it reflected the interests of numerous actors in conserving the area.

### 2.2. Data Collection

In this study, the localities were the social entities selected for understanding the activity systems present in the area. Localities could be ejidos (a communal land tenure system within Mexico), small properties, or private estates (privately-owned ranches). The study was conducted with the free, prior, and informed consent (CLPI) of interviewees and with the permission of municipal and local authorities.

Information was collected through three semi-structured interviews held with 118 key informants between 2015 and 2016 (Table 1). Interviewees were selected following a snowball approach and considering the following criteria: (1) their participation in the original COEPA proposal; (2) their involvement in local decision-making processes (e.g., authorities); and (3) their influence in local

land use (e.g., leaders of local productive and civil organizations). In this sense, sampling followed a stratified approach where all or most key informants (local authorities and leaders) were included.

The first interview was conducted with the people involved in the original proposal for establishing the PA in 2005 (that included government representatives, private owners, and NGO members). This interview had the objective of understanding the limits and the biophysical characteristics of the area originally proposed for the establishment of a PA. The second interview was conducted with key informants of the localities of this area, including ejido and local municipal authorities, private estate owners, and other key informants in terms of providing the information solicited because they had previously been authorities or had specific information. This interview had the objective of obtaining information about activity systems, as well as their perception regarding the delimitation of the area of interest for conservation. Finally, the third interview was conducted with leaders of governmental and non-governmental organizations and institutions involved in the study area in relation to the use of natural resources.

The area of interest for conservation, "Las Cañadas de Sochiapa", was delimited using information obtained in the documents pertaining to the proposal for establishing a PA in 2005 (COEPA 2005) and was complemented with information from interviews with key informants in order to integrate the interests of the local population. The final polygon was traced considering the feedback of local informants, who indicated the sites they considered important for conservation because they were either faunal refuges, relics of native vegetation, or highly fragmented zones susceptible to greater degradation, or because they presented water flows that supplied the region. A subsequent analysis was conducted of the actors related to the area and an analytical categorization was performed in order to clarify their roles and incidence in productive and conservation activities. Thirty-four localities and 42 organizations (i.e., cooperatives, civil associations, governmental institutions) were considered as stakeholders. Modified from Špirić (2016), these were categorized according to their importance, influence, and interest in productive or conservation activities. These categories were: (1) important actors, which were those that had the possibility of developing productive and/or conservation activities in the area, (2) influential actors, which were those that had a cohesive power in decision-making with respect to conservation or productive activities from a regional level and up, and who participated in and had an influence on the formulation and execution of public policies in this regard; and (3) actors with interest, which were those that demonstrated a disposition and will to conserve by conducting conservation actions (i.e., localities with conservation initiatives), or those that depended on productive activities and who promoted the use and management of resources (i.e., localities that depended on productive activities).

Information from interviews 2 and 3 was used to analyze and categorize stakeholders, as well as to understand and describe activity systems. This information was analyzed in a narrative form using descriptive statistics to identify the strengths and weaknesses of the territory, as well as to describe the existing decision-making spaces.

**Table 1.** Type of semi-structured interviews used for data collection. The table includes the number of people interviewed, their percentage representation from the overall target population (in parenthesis), and their role, as well as the information they provided.

| Interviews | Type of Respondents | Number of Interviewees and Percent Representation | Interview Topics | Results |
|---|---|---|---|---|
| 1—COEPA proposal | Participants in the original proposal | 12 (50%) | Location and importance of the polygonal proposed for the PA Prominent aspects of the landscape Redefinition of the area for conservation | Delimitation of "Las Cañadas de Sochiapa" and local description |
| 2—Local actors by municipality | Municipal, community, and *ejido* authorities, Key informants * | 91 interviewees total: Tlaltetela—8 (28%) Totutla—19 (40%) Tenampa—28 (74%) Tlaltetela—15 Totutla—10 Tenampa—11 | Important sites for conservation Activities conducted and resources used: financial, information, and other resources (both material and natural). Aspects of the Territorial, Regulatory and Sectorial dimensions. Rules of access to natural resources and decision-making. Land tenure. Changes in activities/land use over the last 50 years. | Delimitation of "Las Cañadas de Sochiapa" Characterization of activity systems Diversity of actors Local organization and external resources Local management of collective resources |
| 3—Actors involved in land use decision making | leaders and/or representatives of productive and conservation organizations | 13 (45%) | To determine different aspects of organizations: type, activities conducted, scale of action, programs managed, relationships with other organizations and affiliates. | Diversity of actors Local organization and external resources |

* Key informants were specifically selected due to their role in holding information and using the snowball sampling method and therefore it is difficult to estimate their representativeness.

## 3. Results

### 3.1. Delimitation of "Las Cañadas de Sochiapa"

The area delimited as "Las Cañadas de Sochiapa" comprised an area of 21,177.6 ha and integrates parts of the municipalities of Totutla, Tlaltetela, and Tenampa (Figure 1). According to INEGI (2010), there are approximately 10,860 inhabitants, distributed among 12 small properties, 16 ejidos, and five private estates (Table 2).

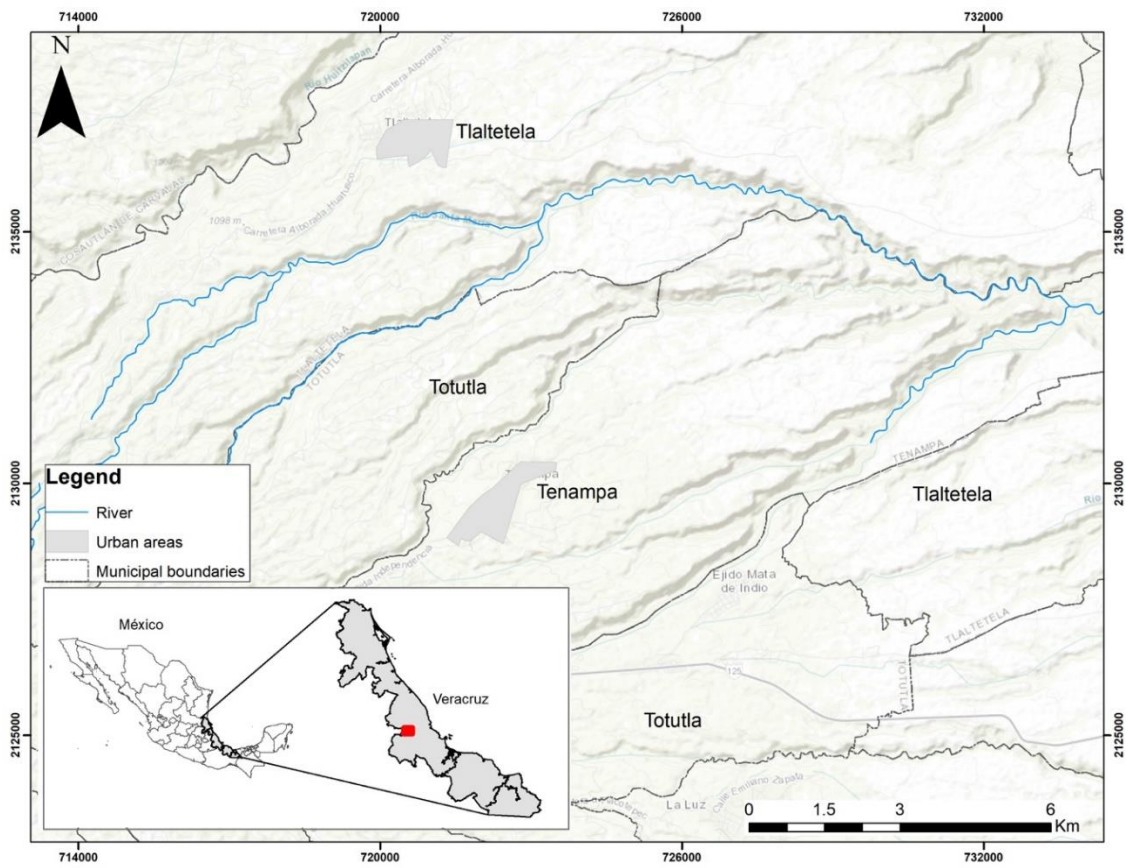

**Figure 1.** Map locating the area of interest for conservation, "Las Cañadas de Sochiapa".

**Table 2.** Characteristics of the population and territory of the municipalities of the area delimited as "Las Cañadas de Sochiapa" *.

| Municipality | No. of Localities of Small Properties | No. of de Ejidos | No. of Private Estates | Total No. of Localities | Total Population |
|---|---|---|---|---|---|
| Tenampa | 5 | 5 | 4 | 14 | 5140 |
| Tlaltetela | 0 | 4 | 0 | 4 | 5716 |
| Totutla | 7 | 7 | 1 | 15 | 11408 |
| Total | 12 | 16 | 5 | 33 | 10856 |

\* Data from the System of Territorial Integration 2010 of INEGI [25].

Delimitation, according to informants, responded to the local area of interest for conservation, mainly due to its natural resources in terms of fauna, flora, and water bodies. The informants reported 83 species of fauna, of which 30 are hunted. Of these, 20 (67%) are consumed in the home or used as pets and the remaining 10 are sold. Of the 10 game species sold, four are sold at a local level and six as hunting trophies, including two exotic species. The interviewees also considered floristic resources as important, identifying 52 species of flora used as shade in the coffee plantations. Of these, 35 (38%)

have more uses, either as firewood or for fruit for self-consumption or sale. In addition, 35 water bodies were reported within the area, including one natural spring.

*3.2. The Landscape of Las Cañadas de Sochiapa*

The landscape of Las Cañadas de Sochiapa is formed by different land uses (seasonal agriculture, coffee agroforestry systems, sugarcane, lime plantations, and induced pastures) distributed around human settlements and with patches of remnant oak forest and other conserved native vegetation found mostly in the deeper gullies. Also, located within the private estates and ejidos, there were zones delimited as conservation areas in which vegetation is naturally recovering or that are intentionally reforested. While these areas do not present mature vegetation as found in the gullies, they constitute areas where secondary vegetation is being recovered.

In descending order in which productive activities present a land cover compatible with conservation goals, the landscape of the three municipalities is dominated firstly by natural vegetation, represented by different forests types, followed by shade-grown coffee (Table 3). This agroforestry system constitutes diversified vegetation areas and much of its biodiversity provides direct and indirect benefits to inhabitants since much of the plant and animal species present have varied uses. However, this agroforestry system is highly threatened due to several factors, including falling market prices, the incidence of pests, and the lack of economic resources to provide maintenance to the crop. Indeed, according to the interviews, there is a tendency to convert coffee systems into monocultures, mainly lime and sugarcane (reported in 50% of the localities) or to abandon coffee areas (reported in 47% of the localities), particularly those found on steep slopes, which are left to regrow as secondary vegetation.

**Table 3.** Area (ha) occupied by the different land and vegetation cover (2014) in "Las Cañadas de Sochiapa". Values obtained from ESA "Land use land cover maps of Antigua river Basin" (2015) [28].

| Land Use Cover | Area (ha) |
|---|---|
| Forests * | 11,277.9 |
| Woody plantations (including shaded coffee) | 4302.4 |
| Agriculture (irrigated and temporal) | 3292.9 |
| Pastures and shrublands (natural and introduced) | 1322.0 |
| Roads | 649.0 |
| Urban areas | 284.2 |
| Water bodies (rivers and lagoons) | 36.6 |
| Bare soil | 12.6 |

* Forest cover includes gallery forests, deciduous forests, tropical montane cloud forest, evergreen forest, and pine-oak forests.

Traditional agriculture is present throughout the region. However, this land use is only present in small areas and plots that are generally much smaller than coffee, sugarcane, or lime plots. Sugarcane is a crop that is found over an extended area, especially in the municipality of Totutla. However, as with coffee, abandonment of this land use was mentioned to be common.

*3.3. Characterization of Activity Systems*

Productive activities conducted in the localities were agricultural production, including subsistence and commercial production, as well as livestock rearing (Table 4), subsistence activities (e.g., gathering, fishing), and activities related to conservation or service provision. In general, localities were shown to practice more than one productive activity.

**Table 4.** Characteristics of the productive activities reported in "Las Cañadas de Sochiapa".

| Activity | Number of Localities | Description | Destination of Production | Products Obtained |
|---|---|---|---|---|
| Shaded coffee | 15 | Coffee grown under the shade of native and exotic trees, some of which produce other products; control of pests and fertilization depends on access to financial aid and size of plot. | Commercial and self-consumption | Besides coffee, macadamia, banana, orange, pepper, among many others. |
| Sun-grown coffee | 14 | Coffee plantation without trees, intensive application of pesticides and fertilizers. | Commercial | No |
| *Milpa* system | 19 | Seasonal with limited use of agrochemicals that depends on financial aid | Commercial and self-consumption | Maize, squash and chili, among others crops and also edible herbs (*quelites*: all edible pants not purposefully cultivated but present in the system) |
| Sugarcane | 11 | Intensified monoculture; depends on acquisition power and contracts with sugar mills | Commercial | Sugarcane |
| Lime | 6 | Intensified monoculture; depends on acquisition power and access to finance | Commercial | Lime |
| Livestock production cattle, sheep and pigs | 5 | The species bred depends on the acquisition power of the producer; cattle is reared in extensive pastures with few trees | Commercial and self-consumption | Livestock |
| Domestic poultry | 4 | Depends on the acquisition power of the producers and access to governmental support | Self-consumption | poultry |

Among agricultural activities, coffee cultivation dominated and was reported in 85% of the localities. Shade-grown coffee was present in only one locality more than sun-grown coffee. Up to 52 tree species can be found in the shade-grown coffee agroforestry systems and as much as 40% of them are used to obtain other goods that can be sold or consumed by families. Governmental support is provided for coffee production (sun and shade grown). Cultivation of maize and beans was reported in 56% of the localities and is generally associated with other crops, such as squash and chili, thus resembling the traditional *milpa* system. While a large part of the maize produced is sold, the beans and other products are for self-consumption. Sugarcane was reported in 32% of the localities and has been cultivated for approximately the last 30 years. The product is bought by the sugar mills of the region, which implies the signing of a contract, access to certain resources for renewing cultivated areas, and credits to cover the cost of production, but also implies the adherence to guidelines related to which variety of sugarcane to sow and types of agrochemicals to use. The sugarcane producers belong to two different unions of producers. Lime has been cultivated in 18% of the localities for approximately five years. This crop has been subject to a strong promotion by a lime producer organization from outside the area, known as the Asociación Citrícola de Martínez de la Torre, which has provided support for sowing and maintenance of the production. Animal raising, whether large-scale or domestic, is complementary in some of the localities, but is not a preponderant activity.

Subsistence activities, including subsistence hunting, fishing, and gathering of non-timber forest products (e.g., mushrooms, herbs, and berries), were reported in 94% of the localities. Eleven localities participated in conservation by having conservation areas, but only one reported conservation as a commercial activity through cinegenic management. This area has a legal management in which owners sell certified access to game animals. However, it is also a conservation area since it conserves the habitat needed for game species and associated fauna, maintaining healthy populations. Finally, in 53% of the localities, there are service providers who work as daily laborers, drivers, and construction workers. In 93% of the localities, remittances are received from family members who work in the United States of America, constituting an important source of income.

### 3.4. Diversity of Actors

The analysis categorized the study localities as actors that are highly important for productive and conservation activities since it is the inhabitants of these localities that make the final decisions. However, the localities are not influential since they have no power of cohesion over other actors or influence over the development of public policies. In terms of interest in conservation, 13 (40%) localities are either involved in conservation in one form or another or manifested their desire to become involved (Table 5). Of these, 11 (33% of the total number of localities) present high interest, since they are involved in conservation initiatives. Six ejidos have delimited areas in which the objective is to conserve the fauna and flora: in five of these, they have designated Private Conservation Areas (PCA) that have a State certificate of protection, and in one, they have designated an area of common use of about 1 ha (less than 1% of the *ejido* area) dedicated to conservation through an internal agreement. These protected areas are solely for natural conservation and no productive activities or extraction of resources are allowed. On the other hand, three localities (9% of the total) have medium interest since they manifested interest in establishing a conservation scheme in the future: two ejidos and one small property. Furthermore, three of the 12 localities that are small properties and two of the five private estates also have PCA (Table 6). One of these PCA in the private estates is the Environmental Management Unit Bellreguart of Sochiapa (UMABS), which conducts sport hunting of native and exotic fauna.

**Table 5.** Localities according to their interest in conservation.

|  | Ejidos (*N* = 16) | Small Properties (*N* = 12) | Private Estates (*N* = 5) | Total (*N* = 33) |
|---|---|---|---|---|
| With a formal PCA | 5 | 3 | 2 | 10 |
| With another scheme of conservation | 1 | 0 | 0 | 1 |
| With no areas of conservation, but manifested interest | 2 | 1 | 0 | 3 |

**Table 6.** Description of the analytical categories in which the organizations involved in productive and conservation activities were placed. Categories adapted from Špiric et al., 2016.

|  | Fundamentals | Direct Promoters | Indirect Promoters | Incidental Promoters |
|---|---|---|---|---|
| Importance | High: objectives include the promotion of agricultural and livestock production activities or conservation; it is contemplated in the local governmental regulations. | High: objectives include the promotion and management of resources for agricultural and livestock production activities. | Medium: objectives include diversification of coffee plantations and expansion of shade coffee production. | Low: objectives include neither the promotion nor management of resources for the field, but they have had this function in the past. |
| Influence | High: since they can develop and/or execute public policies of support to the field or conservation/ reforestation. | No influence: they do not intervene in public policy. | No influence: they do not develop or execute public policies concerning conservation and/or reforestation. | No influence: they do not intervene in public policy concerning agricultural and livestock production activities. |
| Interest | High: they seek to promote and support productive activities and thus manage and/or provide economic resources for this purpose. | High: objectives include the promotion and management of resources for agricultural and livestock production activities | Medium: They have an interest in diversification of coffee plantations and expansion of shade coffee production for which reason they manage resources for reforestation of these sites. | Medium: They show interest in agricultural and livestock production activities, providing support for domestic livestock on a single occasion. |

In total, 42 organizations were recorded in the area, including producer associations, civil society organizations, governmental agencies, or any collective involved in productive or conservation activities. Of the 42 organizations, 22 were involved in activities of conservation and 36 in productive activities, and these were grouped according to importance, influence, and interest in the categories of Fundamentals, Direct promoters, Indirect or secondary promoters, or Incidental promoters (Table 6).

Of the 22 organizations involved in conservation activities, seven (32%) were grouped as Fundamentals and 15 (68%) as Indirect promoters (Table 7). Most of the organizations considered as Fundamentals were governmental agencies that played a critical role in terms of funding and the development and execution of public policies. Two of these were federal, one state, and three at the municipal level. A conservationist Civil Society Organization (CSO) was also included in this category since it contributed resources and training to encourage landowners to certify their properties as PCA. Although the CSO has no influence on public policy or decision-making at a regional level, they are currently regionally influential in promoting the establishment of PCA in several localities.

**Table 7.** Organizations present in each of the categories created, according to their involvement in conservation in "Las Cañadas de Sochiapa". With respect to conservation, no direct or incidental promoters were found.

| Scale of Action | Type | Fundamental | Indirect Promoters |
|---|---|---|---|
| National | Governmental agency | 2 | 0 |
|  | Coffee producer CSO | 0 | 1 |
|  | Campesino CSO | 0 | 1 |
| State | Governmental Agency | 1 | 0 |
|  | NGO | 1 | 0 |
| Municipal | Governmental agency | 3 | 3 |
|  | Coffee producer CSO | 0 | 0 |
| Regional | Campesino CSO | 0 | 2 |
|  | Coffee producer CSO | 0 | 5 |
| Local | Coffee producer CSO | 0 | 3 |
| **Total** |  | **7** | **15** |

The group of Indirect Promoters was formed by governmental agencies that function at the municipal level, and by *campesino* and national or regional coffee producer organizations interested in the diversification of coffee plantations, for which reason they have managed resources for reforestation and conservation activities, such as monitoring the densities of trees and species of fauna.

Only 11% of the 22 organizations with importance, influence, or interest in conservation collaborated among themselves (i.e., developed and/or implemented joint projects). Apart from the conservationist CSO, none reported an integrative or regional project with the objective of conservation or planning of the territory for the coexistence of conservation and productive activities.

In terms of productive activities, almost all of the study localities (30; 88%) were considered to present high interest, since they depend on activities of the primary sector as their principal source of income. Only three reported medium interest since their economy only partly depends on primary sector activities and they mainly depend on activities of the secondary and tertiary sector (Table 8). One of the three is the UMABS and the other two are the municipal centers of Totutla and Tlaltetela.

**Table 8.** Localities according to their participation and interest in productive activities.

|  | Ejidos (*N* = 16) | Small Properties (*N* = 12) | Private Estates (*N* = 5) |
|---|---|---|---|
| Entirely dependent on primary sector activities | 16 | 11 | 3 |
| Main activity not of the primary sector | 0 | 1 | 2 |

The 36 organizations involved in productive activities (86% of the 42) were grouped in turn into Fundamentals, Direct promoters, and Incidental promoters (Table 9). Only two (6%) were considered Fundamentals, both of which were governmental agencies: the Secretaría de Agricultura Ganadería Desarrollo Rural Pesca y Alimentación (SAGARPA) and the Sistema Producto Café de Veracruz. SAGARPA operates at the national level and, in the area, is a promoter of coffee and maize production for self-consumption. As part of SAGARPA, Sistema Producto Café Veracruz seeks to regulate the coffee production chain and address the needs of producers through the joint creation of public policies that support them and promote production.

**Table 9.** Organizations with importance, influence, and/or interest in productive activities that operate in "Las Cañadas de Sochiapa".

| Scale of Action | Type | Fundamental | Direct Promoter | Incidental Promoter |
|---|---|---|---|---|
| National | Governmental Agency | 1 | | |
| | Coffee producer CSO | | | |
| | Sugarcane producer CSO | | 2 | |
| | Campesino CSO | | 2 | |
| State | Governmental Agency | 1 | | |
| | Coffee producer CSO | | 1 | |
| Municipal | Governmental Agency | | 3 | 1 |
| | Livestock producer CSO | | 1 | |
| Regional | Campesino CSO | | 1 | |
| | Lime producer CSO | | 1 | |
| | Agricultural and livestock producer CSO | | 1 | |
| | Sugarcane producer CSO | | 5 | |
| | Coffee producer CSO | | 4 | 2 |
| Local | Coffee producer CSO | | 6 | |
| | Agricultural and livestock producer CSO | | 1 | |
| | Lime producer CSO | | 1 | |
| | Livestock producer CSO | | | 1 |
| | Maize producer CSO | | 1 | |
| | **Total** | 2 | 30 | 4 |

The group of Direct promoters comprised 30 organizations of different types (coffee, livestock and lime producers, governmental and non-governmental bodies, etc., totaling 83% of those involved in productive activities). Most of these (37% of the organizations in this category) are local or regional coffee producer organizations formed by producers that manage resources for coffee plantations. In second place were seven sugarcane producer organizations (19%), which group the producers through contracts with the sugar mills that stockpile regional production. These organizations foment the permanence, intensification, and expansion of the crop through negotiations that guarantee credits and social benefits for producers. The secretaries of Fomento Agropecuario, one in each municipality, are responsible for managing federal resources and for providing information needed by beneficiaries of governmental programs. In one of the three Fomento Agropecuario, it was reported that they promote the production of coffee and the establishment of lime plantations and, in the other two, resources solely support the production of coffee. Two of the campesino organizations are confederations with a presence for more than 10 years in the region, while one is a recently created local CSO (since 2014). Finally, there are two lime, two livestock, two general agricultural and livestock, and one maize producer organizations. The two lime producer organizations are groups of local producers promoted by an organization of lime production from another municipality that exports limes. Despite their short time in existence (since 2003), they have had success in resource management, for which reason they have achieved the maintenance and expansion of the lime crop in the zone. The municipal livestock production association and a local group of livestock producers organized themselves in order to receive support from the municipal government. In general, agricultural and livestock production organizations have the objective of managing resources for the production of

coffee, lime, maize and livestock, etc., for which reason they group a diverse range of producers. The maize producer CSO groups people interested in receiving support in order to continue cultivating this crop. Finally, there is a group of Incidental promoters that includes the municipal DIF (Integral Family Development agency) of Totutla that obtained and distributed resources to a group of women for sheep production that was created for this purpose.

### 3.5. Local Organization and External Resources

The main form in which the people organize themselves to obtain support for productive and conservation activities in the localities is through the organizations described above or through the management of private capital as individuals. For this reason, the organizations are considered important social resources that allow particular activity systems. The number of organizations in each locality varies, but there is an average of three organizations in each. The organizations solicit and receive support from external resources, mainly governmental, which they make available to producers. These are divided into social (pensions and maintenance support, i.e., governmental subsidies such as PROSPERA) and productive (support for agroforestry, agricultural, and livestock production, and conservation/reforestation activities) funding. Other types of non-monetary support are also received, including material resources that constitute inputs for agroforestry or agricultural production and reforestation/conservation activities. Information resources are also considered, which are training programs or workshops related to technical abilities for the improvement of agricultural production or agroforestry activities. Only in three of the seven localities in which there were no organizations were there some type of external resources reported (one social, one productive, and another of conservation/reforestation). On the other hand, in 89% of the 27 localities with organizations, the existence of at least one external resource was reported (Table 10).

**Table 10.** External resources in localities with the presence/absence of organizations. Each locality has more than one resource type.

| | Total No. of Localities | Localities with Social Resources | Localities with Productive Resources | Localities with Conservation/ Reforestation Resources |
|---|---|---|---|---|
| **Localities without organizations** | 7 | 1 | 1 | 1 |
| **Localities with organizations** | 27 | 20 | 22 | 9 |

In total, three social financial resources were recorded, with two social governmental programs present in 30 localities (77%) the most commonly reported. These are programs to assist the elder and marginalized families in general (Programa Pensión para Adultos Mayores and PROSPERA). On the other hand, eleven different types of resources were recorded for productive activities, with the most abundant of these directed toward the production of coffee (82%). The least abundant were those directed towards conservation and/or reforestation (33%).

Access to the different types of resources influences the productive and conservation activities conducted in Las Cañadas de Sochiapa. The most highly productive activity in the study area is the cultivation of coffee, using varieties that are resistant to coffee leaf rust and the fungicides used to combat it, especially in the sun coffee plantations. The varieties of coffee and the fungicides are provided by governmental agencies through subsidies for coffee, sugarcane, and, more recently, lime production.

### 3.6. Local Management of Collective Resources

Notwithstanding the formation of organizations in order to access external resources, there is little organization for the management of natural resources in the localities. In the case of water, it was reported that the local agency responsible for supervising and ensuring the provision of water is

known as the water committee; however, this agency only exists in 44% of the localities. In two other localities, the agency responsible is a state commission and, in one, the municipal department of public works. In the rest of the localities, it is not known how organization for the provision of water takes place. Despite the fact that there are 35 streams that could constitute sources of water for the localities, only two of these watercourses supply 44% of the localities. Of the localities that share their water supply, there are agreements among water committees in only 53% of them. These agreements are established among localities that use gasoline-powered pumps to transport the water from the source to the households and share the same water supply network. This demonstrates the scant organization that exists for the management of natural resources.

Common agreements created within the entities to regulate hunting and tree felling were presented in 75% of the ejidos. However, these agreements are the subject of little discussion during assemblies and, in the particular case of tree felling, the agreements are incongruent with that established in the formal regulations under the law. While state and federal environmental legislation have established that tree felling and hunting is strictly prohibited throughout the territory unless a UMA (Unit of Environmental Management) is established and registered with SEMARNAT, felling and hunting are permitted in the localities and ejidos as long as they are not practiced outside of the landowner's property.

Decision-making spaces are highly related to land tenure. In all ejidos, an assembly is held regularly (at least once every three months). In 67% of the ejidos, a general assembly is also held regularly in which not only the ejido members participate, but also those inhabitants with no agrarian rights and the different committees of the localities, with these being the most inclusive spaces. The most commonly addressed themes in assemblies are those related to the purchase or sale of plots and the problems of invasion of plots by people of the same ejido or elsewhere. Other themes for discussion include the assignation of *faenas* (unpaid community labor) for the maintenance of roads, common spaces, and the water network; establishment of closed seasons; prohibition of hunting and felling; and, on very few occasions, internal regulations. There are no assemblies in the small properties or states unless an announcement is to be made by municipal authorities, which could occur once a year.

## 4. Discussion

The area delimited as Las Cañadas de Sochiapa is a multifunctional landscape that, despite its transformations, presents a diversity of land uses that reflect activity systems that coexist with conservation areas formed by patches of natural vegetation in inaccessible areas and by areas of abandoned rustic coffee plantations. This land use diversity has led to the maintenance of important ecological functions, including biodiversity conservation, water provision, climate regulation, and soil retention [24]. More than 90% of the entities of Las Cañadas de Sochiapa practice agricultural production, for both commercial and self-consumption purposes, in combination with economic income earned from remunerated work in the secondary or tertiary sector, remittances, or subsidies, giving rise to diverse activity systems that vary depending on the productive strategies and the mobilization of available resources.

The varied activity systems that characterize the study area can be explained to a large extent by the diversity of actors and organizations that are present. These actors and organizations have different functions and, for this reason, have differential access to resources, directly influencing the activities they conduct or promote. In this way, as resource managers and decision-makers at multiple scales, they mold the landscape and can tip the balance towards either the possibility of maintaining or damaging the multifunctional character of Las Cañadas de Sochiapa. For example, the presence of organizations is related to a greater procurement of economic and material resources for production that depends on external factors and that affects local interests in conserving the area. It is known that environmental governance, particularly forest conservation and the sustainable use of natural resources, plays a fundamental role in the long-term maintenance of multifunctional landscapes [29]. The strength of multifunctional landscapes as schemes of conservation, in which the diversity of land

uses, or their arrangements, sustains ecosystem functions and, in turn, protects biodiversity [16], lies in the fact that decisions regarding which conservation strategy to adopt fall to the landowners. This is in contrast to that which occurs with the establishment of PA. The presence of organizations in these schemes is key since their success depends on strong environmental governance. However, in the case of Las Cañadas de Sochiapa, despite the notable interest of some actors in maintaining activities that conserve the area, most of the organizations do not necessarily have conservation of the multifunctional landscape as their main axis. Instead, they are more focused on promoting productive activities that do not necessarily go in step with achieving strong environmental governance. Indeed, only 19% (4) of the 22 localities identified as having some interest in conservation are Fundamentals, since they maintain areas dedicated to conservation, while 88% of the 36 organizations focused on productive activities are Fundamentals or Direct promoters of productive activities and do not necessarily follow the principles of SLM [18], failing to contemplate the medium- and long-term maintenance of the multifunctional character of the landscape.

Despite this and considering the criticisms of protectionist models promoted with a focus on PA, which features a lack of consultation with the inhabitants and often generates local conflicts and doubts regarding legitimacy [9], well-orientated multifunctional landscapes can generate strong schemes of environmental governance [30]. The fundamental proposal of multifunctional landscapes is that human intervention should be congruent with the maintenance of ecosystem services [31,32] in order to reconstruct or maintain ecosystems so that they can fulfill objectives of sustainable development and improve human well-being [19]. A way forward is through proclaiming regulations at the regional level that for one part, could favor the conservation of areas and their connectedness in the landscape and for the other, the maintenance or development of SLM practices [18,33].

In this sense, the fact that the economy of the region has been linked for more than 50 years to agroforestry systems, dominated particularly by shade-grown coffee, is a positive aspect that allows the conservation of important characteristics of the ecosystems that sustain its functions. The complex vegetation structure of agroforestry systems can closely resemble that of native forests, enabling them to host many species [34–37]. Moreover, these diversified systems have the possibility of improving productive activities while promoting biodiversity conservation [38] and could act to mitigate changes in temperature and precipitation [39], thus contributing to the formation of multifunctional landscapes. It has been stated that functional biodiversity in agroforestry systems is benefited by the presence and diversity of shade trees [40,41]. In Las Cañadas de Sochiapa, there is a reported diversity of up to 52 species of trees used as shade in coffee plantations that also provide sub-products for self-consumption. However, the permanence of these coffee production systems in the area is at risk due to the fluctuation in market prices, the incidence of pests, and conversion into monocultures.

One key aspect for effective governance is the existence of relationships among decision-makers at different levels, particularly when there are common use or shared resources, as well as recognition of agreements and rules at different scales. For example, in a context in which there is a shared resource, the users of this resource should ideally establish an agreement for its management. The contrary, as indicated by Bodin and Tengo (2017) [42], represents a threat to the permanence of this resource. Participation, collaboration, and learning are processes that must take place to ensure effective governance [29]. The failed effort to establish a PA in Las Cañadas de Sochiapa is a clear example of the importance of common agreements and objectives among stakeholders for proceeding with a conservation initiative.

Nevertheless, despite the links that exist among decision-makers with an interest in conservation, current environmental governance is characterized by a lack of common agreements and communication between the users and the regulations for hunting and tree felling. In the case of water use, one indicator that the principle is not being accomplished is that no communication or agreements exist among localities that use the same water bodies. On the other hand, agreements with respect to tree felling and hunting are rarely discussed since these resources are not perceived as common, but rather as resources that belong to particular owners (family plots). This lack of rules and

agreements is symptomatic of inadaptability because of the lack of clear and collectively established rules regarding access to and distribution of resources [43].

In addition to the scant discussion of common agreements, the lack of inclusive spaces for decision-making (general assemblies only exist in 12% of the localities) implies that the exchange of information and conciliation of interests may be a challenge, constituting an obstacle to the development of strategies of adaptive management in which conflicts among actors can be resolved and the multifunctional landscapes maintained, as indicated by Hodbod and collaborators (2016) [44]. The long-term maintenance of Las Cañadas de Sochiapa as a multifunctional landscape depends on the continuation and expansion of inclusive spaces for decision-making in which different stakeholders, such as landowners; producers; and the local, municipal, and community authorities, can engage in dialogue and negotiate to foster conservation and the continuity of agroforestry systems, such as shade-grown coffee systems. One determinant factor for the permanence of multifunctional landscapes is collaboration and the exchange of information in order to reach agreements. In the case of Las Cañadas de Sochiapa, there is scant collaboration in terms of conservation (only 11% of the organizations interested in conservation collaborate among themselves) and only one of these bodies reported an integrative project at a regional level. There is no initiative that seeks the territorial regulation, at a small or large scale, that could achieve conciliation of conservation with productive activities. A potential solution to achieve a common framework to design a local multifunctional landscape is through identifying the influential, relevant, and interested stakeholders, both regarding conservation and productive activities, and promoting their interaction. Among the principles of developing landscape approaches to conservation according to [33] is to develop negotiation skills among stakeholders through fostering trust and clarifying rights and responsibilities [33]. Understanding the motivations that different stakeholders have on the landscape can aid in building spaces for negotiation among actors. Experiences in integrated environmental policy initiatives can provide guidance as to how to coordinate landscape planning efforts and promote capacity building [45–47], aiding in the maintenance of multifunctional landscapes. Efforts such as Integrated Environmental modeling, for example, in which other actors such as scientists derive information of relevance for landscape planning, can help provide outputs that can make sense for people involved in the management at regional levels [46].

In the current context in which environmental change demands a high capacity for response, it is paradoxical that agricultural intensification continues to increase when the consequence is a reduced resilience of systems that are thus more vulnerable to disturbance [39,48,49]. Contrary to the general perception among coffee producers of Las Cañadas de Sochiapa, it has been demonstrated that the short-term increase in coffee yield produced by the removal of shade acts to reduce the long-term resistance and resilience of the system since it becomes difficult to control pests and the system becomes more vulnerable to the impact of climate change, as has also been reported for agroforestry systems of cocoa [50]. For these reasons, the conversion of coffee plantations puts the multifunctional character of the area at risk in the medium- and long-term [51,52]. In Las Cañadas de Sochiapa in particular, we believe that the development of an integrative project that enables the maintenance of conservation spaces and agroforestry systems, as well as sustainable resource management, would constitute a suitable strategy to ensure the multifunctionality of the landscape. Such a landscape vision permits the simultaneous addressing of multiple objectives across different scales.

The use of spatial planning to integrate environmental policies and agendas has great potential for achieving multifunctional landscapes. As proposed by Votsi et al. [47], one way to assess how different goals can be achieved in the same territory is through contrasting the execution of different policies in similar geographical areas and understanding if they mutually reinforce or weaken each other. This can help to identify key factors to ensure the effectiveness of multifunctional landscapes.

## 5. Conclusions

Despite the threats present in multifunctional landscapes such as Las Cañadas de Sochiapa, including land use change fomented by market demands, the fall in prices of agroforestry crops, and the lack of robust environmental governance, multifunctional landscapes constitute a viable conservation alternative to the vertical establishment of PA since they represent strategies with a fundamental participative component that could potentially seek to reconcile interests for the longer term conservation of ecosystem functions [16]. In this regard, the presence of local organizations and spaces for conciliation are key factors to successfully achieve conservation, even if their focus is not conservation per se. Collaboration, participation, and learning among actors at different landscape scales are the processes that determine natural resource management and environmental governance and are thus fundamental to the planning and conservation of the landscape. As such, bringing together the common interests of stakeholders regarding resource management can be essential for achieving multifunctional landscapes as conservation schemes. In these, landowners, producers; and local, municipal, and community authorities, as well as stakeholders at other levels, are favored by contexts of dialogue and negotiation, leading to strong environmental governance. Ideally, a multifunctional landscape as a conservation alternative would maintain a high-quality matrix in which production occurs mostly through agroecosystems that include SLM practices, allowing for the preservation of natural patches of vegetation with a high degree of connectedness, economically benefiting stakeholders and the region. Integrated environmental policy planning can be the means for coordinating landscape planning efforts for maintaining multifunctional landscapes.

**Author Contributions:** Conceptualization, J.R.-C., L.P.-B. and M.B.-M.; methodology, J.R.-C., L.P.-B. and M.B.-M.; validation, L.P.-B. and M.B.-M.; formal analysis, J.R.-C. and L.P.-B.; investigation, J.R.-C.; resources, L.P.-B. and M.B.-M.; data curation, J.R.-C., L.P.-B. and M.B.-M.; writing—original draft preparation, J.R.-C.; writing—review and editing, J.R.-C., L.P.-B. and M.B.-M.; visualization, J.R.-C.; supervision, L.P.-B. and M.B.-M.; project administration, L.P.-B.; funding acquisition, L.P.-B. and M.B.-M.

**Funding:** Julia Ros-Cuéllar was supported by Consejo Nacional de Ciencia y Tecnología (CONACyT-México 338144) during her master's in science studies in which this research was conducted.

**Acknowledgments:** We thank the Consejo Nacional de Ciencia y Tecnología (CONACYT) for providing a scholarship to the first author and to the Instituto de Ecología, A. C. (INECOL) for providing the support for conducting this research. We would like to thank Octavio Pérez Maqueo, Eduardo García-Frapolli, Emma Villseñor, Sherie Rae Simms, Alfonso Langle, Estrella Chevez Martín del Campo, and Juan José Von Thaden, for all the support they provided for doing field work and for the interview design. We also thank the families and informants from Tenampa, Tlaltetela, Totutla, and the UMA Bellreguart de Sochiapa who shared their knowledge and time with us. The article benefited greatly from three anonymous reviewers.

**Conflicts of Interest:** The authors declare no conflict of interest.

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
