# Peer review of "Can Multifunctional Landscapes Become Effective Conservation Strategies? Challenges and Opportunities From a Mexican Case Study"

_land, doi:10.3390/land8010006_

Round 1
Reviewer 1 Report
In this paper it is analyzed whether the Multifunctional Landscape (ML) concept was an operative alternative to Protected Areas (PA) within an area of interest for conservation in Veracruz, Mexico.
The case study proposed in this article was approached through a conceptual framework based on 3 points: delimit the area of interest for conservation as ML, understanding the role played by different actors in this landscape and identifying strengths and weaknesses for developing future strategies allowing ML scenarios as alternatives to PA.
Why the establishment of a state level PA failed?
What kind of lessons could be learned from this failure and used in ML?
From this article (Introduction section), ML seems to be defined someway similar with sustainable land management concept (SLM). SLM is defined as the use of land resources, including soils, water, animals and plants, for the production of goods to meet changing human needs, while simultaneously ensuring the long-term productive potential of these resources and the maintenance of their environmental functions (Motavalli et al., 2013).
The article presented very interesting data resulted from the applied questionnaires. However, introducing ML as a potential sustainable alternative solution to PA is not convincing. A high number of interests covering a relative small region will be difficult to be managed for assuring wellbeing of human meanwhile with ensuring natural resources conservation.
The results of this article emphasize the lack of links/ relationships between different key stakeholders in supporting ML as a key alternative to PA. Unfortunately, the article doesn’t provide any answer to this problem.
Establishing some specific SLM regulations at regional level can be a better solution than adopting the ML concept. Introducing SLM in the ML analysis will definitely bring an added value.
As a conclusion, this article needs a revision especially regarding section 4 (Discussions) and 5 (Conclusions).
Author Response
Reviewer 1 |
Dear Reviewer, In the next lines we provide answers to all of your comments as well as directions to find changes in the manuscript. |
Comment Why the establishment of a state level PA failed? Answer We included in the introduction a more detail explanation of how the area was proposed as a PA in 2005 and why it failed. Lines 95-106. |
Comment What kind of lessons could be learned from this failure and used in ML? Answer In the discussion, the lessons learned from that previous experience (mostly lack of coordination among stakeholders) are included to exemplify the importance of common agreements for landscape planning (lines 515-517). |
Comment From this article (Introduction section), ML seems to be defined someway similar with sustainable land management concept (SLM). SLM is defined as the use of land resources, including soils, water, animals and plants, for the production of goods to meet changing human needs, while simultaneously ensuring the long-term productive potential of these resources and the maintenance of their environmental functions (Motavalli et al., 2013). Response SLM and ML are similar concept, although they occur at different scales. While SLM refers to a management unit, ML refers to a landscape perspective composed of many SLM units. Actually, introducing the SLM in the introduction (Lines 60 to 65) helped to integrate fully the idea of ML as a landscape where conservation areas are immersed in a high-quality matrix composed of SLM units. |
Comment The article presented very interesting data resulted from the applied questionnaires. However, introducing ML as a potential sustainable alternative solution to PA is not convincing. A high number of interests covering a relative small region will be difficult to be managed for assuring wellbeing of human meanwhile with ensuring natural resources conservation. Answer In the discussion section we provide arguments sustained by the literature of how landscape planning can reconcile agriculture with conservation (see Lines 492 to 498). The concept of SLM was very useful for broadening our vision and the explanation of what we were intended to say. Also, comments by other reviewers, such as considering Integrated environmental policy initiatives were useful for operationalizing the idea of multifunctional landscapes as spaces that lead to the reconciliation human wellbeing while ensuring natural resources conservation (see paragraph 530 to 557). |
Comment The results of this article emphasize the lack of links/ relationships between different key stakeholders in supporting ML as a key alternative to PA. Unfortunately, the article doesn’t provide any answer to this problem. Answer We included a paragraph attending this concern in which we explain a potential solution to this problem by seeking a common framework in which negotiation spaces can be developed through approaches such as integrated environmental policy initiatives. For this we cite several references that specify the importance of negotiation among actors and building trust through clarifying rights and responsibilities (Parson, 1995. Laniak et al., 2013, and Sayer et al., 2013). Lines 530-557 |
Comment Establishing some specific SLM regulations at regional level can be a better solution than adopting the ML concept. Introducing SLM in the ML analysis will definitely bring an added value. Answer We consider as explained before that SLM and ML are not similar concept but rather that developing SLM management units is a way of strengthening ML. We tried to make our arguments stronger of the viability of producing ML by introducing the importance of developing SLM units in the landscape, and the importance of having regulations at the regional level regarding the development of SLM. We mention this in the discussion in lines 496-498. Here we also make reference of Sayer et al., 2014 in order to introduce the landscape level concept. In line 487 in referring to weaknesses of the current system we also mention a lack of SLM vision. |
Comment As a conclusion, this article needs a revision especially regarding section 4 (Discussions) and 5 (Conclusions). Answer We have revised both sections following all reviewers’ comments. We have also included more references to sustain our arguments and thought more thoroughly the conclusions. |
Reviewer 2 Report
This manuscript describes an interesting research about the conservation potential of Multifuctional Landscapes. Though the research contains a high degree of novelty, revisions should be made in order for the paper to be published. More specifically:
Citations describing what has already been mentioned in the integrated environmenmtal policy initiatives should be added to the manuscript. For example,
Votsi, N. E. P., Kallimanis, A. S., Mazaris, A. D., & Pantis, J. D. (2014). Integrating environmental policies towards a network of protected and quiet areas. Environmental conservation, 41(4), 321-329.
Parson, E. A. (1995). Integrated assessment and environmental policy making: in pursuit of usefulness. Energy Policy, 23(4-5), 463-475.
Laniak, G. F., Olchin, G., Goodall, J., Voinov, A., Hill, M., Glynn, P., ... & Peckham, S. (2013). Integrated environmental modeling: a vision and roadmap for the future. Environmental Modelling & Software, 39, 3-23.
Results should be presented in a form to answer the goals of the study in a more elusive way.
More attention should be paid in the conclusion section so as to answer to paper's objectives but also to delimit the framework of future steps.
Author Response
Answers to comments of Reviewer 2
Comment This manuscript describes an interesting research about the conservation potential of Multifuctional Landscapes. Though the research contains a high degree of novelty, revisions should be made in order for the paper to be published. More specifically: Citations describing what has already been mentioned in regard to integrated environmental planning should be added to the manuscript. For example, Votsi, N. E. P., Kallimanis, A. S., Mazaris, A. D., & Pantis, J. D. (2014). Integrating environmental policies towards a network of protected and quiet areas. Environmental conservation, 41(4), 321-329. Parson, E. A. (1995). Integrated assessment and environmental policy making: in pursuit of usefulness. Energy Policy, 23(4-5), 463-475. Laniak, G. F., Olchin, G., Goodall, J., Voinov, A., Hill, M., Glynn, P., ... & Peckham, S. (2013). Integrated environmental modeling: a vision and roadmap for the future. Environmental Modelling & Software, 39, 3-23. Answer We have included all the references in the discussion section and have use them to settle the frame for next steps and recommendations (Lines 545-557 and 571-575). The readings were very useful for providing an idea of how to operationalize multifunctional landscapes as conservation schemes and introducing the idea of integrated environmental planning at landscape level. We hope to have integrated a more complex idea of what we are trying to say. |
Comment Results should be presented in a form to answer the goals of the study in a more elusive way. Answer Following reviewers’ comments, we detailed various aspects of the results. In particular, we have given more information on the land cover of the site, including a new table (Section 3.2.); clarified the extent of conservation in one ejido (section 3.4). In addition, we modified table 1 and made specific references to what kind of results provided each interview type. |
Comment More attention should be paid in the conclusion section so as to answer to paper's objectives but also to delimit the framework of future steps. Answer We have revised this section and included more relevant points derived from our results. In particular, the importance of including diverse interests to successfully achieve conservation through multifunctional landscapes and also the importance of creating spaces for the different actors to communicate for landscape planning. We introduced the idea of integrated environmental policy planning for operationalizing multifunctional landscapes as conservation schemes. |
Reviewer 3 Report
General remark
Some references/discussion should be done in relationship with social-ecological systems (a bit more away than socioecological approximation, Line 73), and about land sharing/land sparing perspective. Adding some references to these questions could improve the background of land strategies conservation, the aim (one of them) of this manuscript.
Data collection section: Through the text or in Table 1 the representative percentage of participants/locals/leaders/localities interviewed/studied, in relationship with all of them belong to the study zone, should be showed (explicit the representative degree of interviews).
Results section text may be more explicit. This section should showed the results obtained in direct relationship with Table 1- second column item “information obtained”. Rewriting recommendation.
The percentage of land occupied by each main crops/wild vegetation (and their spatial distribution) should be showed in order to take a general picture of the “intensification” of the landscape. This methodological approximation may be derived into a better understanding of land protection fact under the present/future situation. If this information is available, it will be easier do some prediction about landscape evolution after the new monoculture implementation tendence.
Partial remarks
Line 236-37. How much is the % of income from the “USA workers” in relationship with family total income. This figures could explain some personal/family positions about conservation/mono crop-cultures preferences.
Line 241. “However, the localities are not influential...” Do you have some figures in order to support this statement?
Line 248. “...about 1 ha dedicated to conservation...” How much (%) is this area in relationship with the whole area?. This 1 ha is translated in a low/medium/high crop production reduction (less quantities)? Perhaps this consideration may be useful, again, in order to explain personal/community position about preservation/crop intensification dilemma.
Author Response
Answers to comments of Reviewer 3
|
Comment Some references/discussion should be done in relationship with social-ecological systems (a bit more away than socioecological approximation, Line 73), and about land sharing/land sparing perspective. Adding some references to these questions could improve the background of land strategies conservation, the aim (one of them) of this manuscript. Answer In the introduction, we make a more specific explanation of what we mean by socioecological approximation by explaining the concept of socioecological systems as complex systems and citing Berkes and Folke 1998 and Ostrom 2009 (lines 85 – 89). Also, we include in the introduction a further explanation of the multifunctional landscape within the discussion of land sharing/sparing models (lines 54-60). We cite Perfecto and Vandermeer 2010 and 2012. |
Comment Data collection section: Through the text or in Table 1 the representative percentage of participants/locals/leaders/localities interviewed/studied, in relationship with all of them belong to the study zone, should be showed (explicit the representative degree of interviews). Answer In the text, we have explained with more detail how interviewees were selected (lines 146-151). Since we were specifically interested in the information from key informants, we followed a snowball technique. Also, in table 1, that we modified, we have included two more new columns. In one of them we added the percentage of respondents with regards to the universe of possible respondents and stratified the information by type of actor. |
Comment Results section text may be more explicit. This section should showed the results obtained in direct relationship with Table 1-second column item “information obtained”. Rewriting recommendation. Answer We have modified Table 1 and one of the columns we added was to make reference to the type of results obtained from each interview type in order to make more explicit what was the information obtained used for. |
Comment The percentage of land occupied by each main crops/wild vegetation (and their spatial distribution) should be showed in order to take a general picture of the “intensification” of the landscape. This methodological approximation may be derived into a better understanding of land protection fact under the present/future situation. If this information is available, it will be easier do some prediction about landscape evolution after the new monoculture implementation tendency. Answer We have included a new table (number 3) showing the area (ha) that the different land covers occupied in the region in 2014. We referenced it in the text in the results section 3.2. The landscape of Las Cañadas de Sochiapa (lines 224-226) citing the new table. The information was derived from the European Space Agency, Earth Observation for Development, World Bank, Brockmann Consult y GeoVille. 2014. Land use land cover maps of Antigua river Basin. Coastal Watersheds in Mexico. This source included information for the area discussed. |
Comment Line 236-37. How much is the % of income from the “USA workers” in relationship with family total income. These figures could explain some personal/family positions about conservation/mono crop-cultures preferences. Answer We did not obtain this information nor explored the relation of income with conservation preferences or with activity systems |
Comment Line 241. “However, the localities are not influential...” Do you have some figures in order to support this statement? Answer As written in data collection section, we define influential actors as “those that had a cohesive power in decision-making with respect to conservation or productive activities from a regional level and up, and who participated in and had influence on the formulation and execution of public policies in this regard” (lines 184-186). In this sense, localities do not have a cohesive power in decision-making at upper levels or are influential in the formulation of public policy. No figures to support the statement. |
Comment Line 248. “...about 1 ha dedicated to conservation...” How much (%) is this area in relationship with the whole area? This 1 ha is translated in a low/medium/high crop production reduction (less quantities)? Perhaps this consideration may be useful, again, in order to explain personal/community position about preservation/crop intensification dilemma. Answer The specific sentence the reviewer points out only refers to the area devoted to conservation in one ejido (the only ejido that has no formal APC). The ejido´s total area is about 218 ha; therefore the 1 ha only represents <1% devoted to conservation where no resource management or productive activities are conducted. We have included that in the text (lines 290 to 293 in section 3.4 Diversity of actors). |
Round 2
Reviewer 1 Report
The authors succeeded to answer reviewer requests after first evaluation. Given the improved form of the manuscript, it can be published now in the journal.
Author Response
The reviewer indicated that the comments in the revision were addressed successfully. We corrected the grammar an punctuation errors as much as we could and made changes to the abstract as suggested by the editor.